# Purification of Sewage Wastewater though Sand Column Filter for Lessening of Heavy Metals Accumulation in Lettuce, Carrot, and Cauliflower

**Safina Naz** [1], **Muhammad Akbar Anjum** [1], **Bushra Sadiq** [2], **Riaz Ahmad** [3,*], **Muhammad Ahsan Altaf** [4,*], **Mohamed A. El-Sheikh** [5] **and Awais Shakoor** [6]

1   Department of Horticulture, Faculty of Agricultural Sciences & Technology, Bahauddin Zakariya University, Multan 60800, Pakistan
2   Faculty of Agriculture and Environmental Science, Islamia University of Bahawalpur, Bahawalpur 63100, Pakistan
3   Department of Horticulture, The University of Agriculture, Dera Ismail Khan 29120, Pakistan
4   College of Horticulture, Hainan University, Haikou 570228, China
5   Botany & Microbiology Department, College of Science, King Saud University, P.O. Box 2455, Riyadh 11452, Saudi Arabia
6   Teagasc, Environment, Soils and Land Use Department, Johnstown Castle, Co., Y35 Y521 Wexford, Ireland
*   Correspondence: riaz.ahmad@uad.edu.pk (R.A.); ahsanaltaf8812@gmail.com or ahsanaltaf@hainanu.edu.cn (M.A.A.)

**Abstract:** Sewage wastewater is one of the richest sources of mineral nutrients contributing toward plant growth and yield. However, the accumulation of heavy metals in the edible parts of vegetables and fruits can be dangerous to life. The current research aimed to evaluate the performance of a sand column filter for the elimination of heavy metals from sewage wastewater applied to selected vegetables. The contents of heavy metals, i.e., $Pb^{+2}$, $Ni^{+2}$, $Cu^{+2}$, and $Fe^{+2}$, were estimated to be higher in untreated sewage wastewater than in treated water. The number of leaves, fresh and dry weights of leaves, roots, and total biomass of lettuce, carrot, and cauliflower were improved due to the irrigation of unfiltered sewage wastewater compared to sewage wastewater. The curd diameter, fresh and dry weights of curd, stem fresh weight of cauliflower, and the root length and diameter of carrot increased after irrigation with the unfiltered sewage wastewater in comparison to the treated sewage wastewater. The accumulation of heavy metals, i.e., $Pb^{+2}$, $Ni^{+2}$, $Cu^{+2}$, and $Fe^{+2}$, was higher in the roots, leaves, and edible parts of the selected vegetables. In the present study, the $Cd^{+2}$ and $Cr^{+2}$ concentrations were not affected by the filtration process through a sand column filter. Conclusively, filtration through a sand column filter is effective for the removal of heavy metals from sewage wastewater used to irrigate agricultural land.

**Keywords:** growth and yield; heavy metal concentration; plant biomass; reclamation of sewage wastewater



## 1. Introduction

Freshwater is a unique resource with imperative qualities. Due to increased demand, the availability of clean water has reduced approximately three-fold in South Asia, Africa, and the Middle East since the 1950s [1]. Poor management in the distribution of irrigation systems, ground water with a salty nature, low rainfall, and the rapid increase in the population are the major causes of water deficit conditions. This situation is leading towards the use of untreated city wastewater for the irrigation of agricultural land. The utilization of sewage wastewater is becoming more popular throughout the developing world [2], particularly in peri-urban areas of Multan, Pakistan [3].

Wastewater is rich in the organic and inorganic nutrients necessary for the sufficient growth and yield of plants [4]. Farmers frequently use wastewater for their farming

purposes in those areas located around industrial zones and cities. Farmers consider that this is an inexpensive and valuable source of minerals. Hence, farmers use it as suitable alternative to fertilizers for plant growth [5]. Macronutrients, i.e., N, P, and K; micronutrients, i.e., Fe, Cu, and Zn; and a considerable amount of organic matter are vital nutrients present in sewage wastewater that contribute to plant growth and production [6]. The presence of macro- and micronutrients in sewage wastewater has been shown to enhance growth and yield in corn [7] and mulberry [8]. Similarly, increased growth and yield was also recorded in chili [9].

An excess of heavy metals in soils is mainly due to sewage wastewater irrigation [10]. Excessive heavy metals deposition in agricultural land results in an excess of heavy metal accumulation in different plant parts. This accumulation is a serious threat to food safety around the globe [11]. The recommended safe concentrations of heavy metals for human consumption are $Fe^{+2}$ 150 mg $kg^{-1}$, $Pb^{+2}$ 2 mg $kg^{-1}$, $Cu^{+2}$ 10 mg $kg^{-1}$, $Ni^{+2}$ 10 mg $kg^{-1}$, $Cd^{+2}$ 0.02 mg $kg^{-1}$, and $Cr^{+2}$ 1.3 mg $kg^{-1}$ [12]. The consumption of fruits and vegetables contaminated by heavy metals is a serious danger to human life [13]. These metals are very toxic and non-biodegradable, and even their ingestion at the parts-per-billion level causes numerous diseases [14]. Heavy metals have carcinogenic, teratogenic, neurotoxic, incurable, and mutagenic concerns [15], have known adverse effects on blood acidity, and can lead to kidney damage, cancer, and retardation [8]. These situations might be shocking for vegetarians because their main food component is fresh vegetables.

Pakistan's irrigation system is mainly based on the canal water of the Indus Basin [8]. Canal water is not sufficient to fulfill the farming requirements of country farmers due to the increased population and different environmental fluctuations. Irrigation with sewage wastewater is a very common practice in Pakistan. However, no serious actions have been taken to eradicate these dangerous concerns. Different management practices, e.g., slow sand filters [16], thick tire chips [17], rubber chips [18], and fine medial filters [19], are used for the elimination of heavy metals from sewage wastewater. However, their application is still limited around the world.

For the reduction of heavy metals, sand-based columns are used in tomato and okra crops [20]. However, sand-based columns are not used for lettuce, cauliflower, or carrot. Therefore, the present work was carried out to evaluate the capability of sand columns for sewage wastewater filtration to lessen the concentration of heavy metals and also to evaluate the effects of unfiltered and filtered sewage wastewater on the growth, yield, and heavy metal buildup in the leaves, roots, and edible parts of the selected vegetables.

## 2. Materials and Methods

### 2.1. Experimental Details

The present research was carried out for the evaluation of the growth, yield, and heavy metals in the selected vegetables, i.e., lettuce, carrot, and cauliflower. The vegetables were grown under filtered and unfiltered sewage wastewater for two years (2014 and 2015) in the specific research site of the Department of Horticulture, Bahauddin Zakariya University (BZU), Multan. Earthen pots were filled with 5 kg silt loam soil below the corner to ensure easy watering. The main disposal unit of BZU campus was the source of the sewage wastewater for the treatments.

### 2.2. Preparation of Sand-Based Column

For the sewage wastewater filtration, the sand-based column contained mesh-sized sieves (0.5 mm). An iron sheet was placed inside and an outlet was installed at the bottom. The column was filled with sand and the sewage wastewater was passed through it. After that, the filtered water was collected in a container and the sand was changed after one filtration for the higher efficacy of the column. Afterward, the treatment—either filtered or unfiltered sewage wastewater—was applied to the pots of the selected vegetables. The present research adopted a completely randomized design (CRD) including three repeats, and each repeat comprised ten pots. The cultural practices for each crop were similar.

### 2.3. Growth and Yield Traits

At the maturity stage, the growth and yield of the selected vegetables were measured from the treated and untreated plants. The leaves were counted from each treatment. The fresh and dry weights of the leaves, roots, plant biomass, and curd were taken using a digital weighing balance. The curd diameter, root length, and root diameter were measured using Vernier calipers.

### 2.4. Determination of Heavy Metals

2.4.1. Chemical Preparation

For the extraction of the heavy metals, glassware was soaked for 12 h in a nitric acid solution of 10%. For the estimation of the standard value at 1000 mg $L^{-1}$, $Ni^{+2}$, $Cu^{+2}$, $Cd^{+2}$, $Pb^{+2}$, $Fe^{+2}$, and $Cr^{+2}$ standards were bought from Merck and Co.

2.4.2. Pre-Treatment of Sewage Wastewater before and after Filtration

The water samples, i.e., filtered and unfiltered, were collected from the disposal unit and stored in airtight plastic bottles. According to the method of Singh [21], approximately 5 mL of $HNO_3$ was poured into the plastic bottles to prevent any reaction of the heavy metals with the plastic bottles during storage.

2.4.3. Digestion of Plant Samples for Heavy Metal Extraction

The different plant parts, i.e., the roots, leaves, and edible parts, were harvested for the determination of heavy metals from the unfiltered sewage wastewater-treated (control) and the filtered sewage wastewater-treated plants. Distilled water was used for washing the plant parts, which were dried by spreading them out in a clean place for 60 min. After that, the plant parts were divided into small pieces and put into the oven, where they were dried until a constant weight was achieved. The dried samples were finely ground into a powder and stored in polyethylene zipper bags under ambient conditions for further downstream analysis. Approximately 0.5 g of the ground powder from each treatment was subjected to acidic digestion in a 3:1 ratio containing 15 mL of $HNO_3$ and 5 mL of $HClO_4$. After that, the samples were placed on a hot plate at 80 °C. The samples were heated until the appearance of a colorless material. This material was collected in beakers for future downstream analyses, as per the described method of Singh [21].

2.4.4. Extraction of Heavy Metals from Plant and Water Samples

The wet digestion method was used for the extraction of heavy metals from the water samples. After the digestion of water and plant samples of the selected vegetables, the samples were filtered through Whatman No. 42 filter paper. The filtered samples and 25 mL of distilled water were homogenized and stored in plastic bottles. The previously described method of [21] was used for the extraction of heavy metals from the collected samples and identified through atomic absorption spectroscopy as well as a photometer.

### 2.5. Statistical Analysis

The collected data were evaluated using Statistix 8.1 as described in [22] under a two-way factorial design, i.e., years and sewage wastewater treatment. However, the interaction between the years and sewage wastewater treatment was found to be non-significant, and therefore is not discussed in the Section 3. The means of the sewage wastewater treatments were separated using LSD at the 0.05 level of probability.

## 3. Results

### 3.1. Sewage Wastewater Characteristics before and after Filtration Used in the Experiment

The physiochemical properties, i.e., pH, EC, $HCO_3$, RSC, Ca + Mg, Sulfate, Na, and Cl, of the sewage wastewater before and after filtration were used (Table 1).

**Table 1.** Physiochemical properties of sewage wastewater before and after filtration used in the experiment.

| Sewage Wastewater | pH | EC (dSm$^{-1}$) | HCO$_3$ | RSC | Ca + Mg (meq L$^{-1}$) | Sulfate (meq L$^{-1}$) | Na (meqL$^{-1}$) | Cl (meqL$^{-1}$) |
|---|---|---|---|---|---|---|---|---|
| Unfiltered | 7.3 | 0.183 | 13 | 4.4 | 9.6 | 1.09 | 8.79 | 5.02 |
| Filtered | 7.5 | 0.391 | 10 | 3.5 | 7.5 | 0.79 | 6.91 | 4.5 |

### 3.2. Heavy Metal Content in Unfiltered and Filtered Sewage Wastewater Samples

The unfiltered sewage wastewater revealed significantly higher Pb, Ni, Cu, Cd, Fe, and Cr levels compared to the filtered sewage wastewater (Table 2). According to WHO/FAO and Naz et al. (2021), all of the studied heavy metals were found to be high in unfiltered water, while lower heavy metals were recorded in filtered water. A more significant reduction in the heavy metal content was observed in the filtered water than unfiltered sewage wastewater (Table 2).

**Table 2.** Heavy metal content (mg L$^{-1}$) in sewage wastewater before and after filtration used in the experiment.

| Sewage Wastewater | Pb$^{+2}$ | Ni$^{+2}$ | Cu$^{+2}$ | Cd$^{+2}$ | Fe$^{+2}$ | Cr$^{+2}$ | TDS | Hardness |
|---|---|---|---|---|---|---|---|---|
| Unfiltered | 2.48 a | 1.96 a | 3.26 a | 1.26 a | 8.76 a | 0.46 a | 200 | 700 |
| Filtered | 1.71 b | 1.35 b | 2.58 b | 0.83 b | 6.30 b | 0.32 b | 160 | 570 |
| * WHO/FAO | 5.0 | 0.2 | 0.2 | 0.2 | 5.0 | 0.1 | 70 | 500 |

Note(s): * = Permissible limits of heavy metals. The treatment mean of three samples with different letters showed a significant difference at $p \leq 0.05$ (LSD test).

### 3.3. Growth and Yield of Spinach, Carrot, and Cauliflower

The leaf number, leaves, roots, and total biomass of fresh and dry weights were significantly greater among the unfiltered sewage wastewater-treated plants, while these traits were significantly lower in the filtered sewage wastewater-treated plants of the selected vegetables, i.e., spinach, carrot, and cauliflower (Table 3). However, the number of leaves, the fresh and dry weights of the leaves and roots, and the total biomass were not affected by year (Table 3). The fresh weight, dry weight, and diameter of curd, as well as the stem fresh weight were found to be higher in the unfiltered sewage wastewater-treated samples, while the curd diameter, fresh and dry weights of curd, and stem fresh weight were found to be lower in the filtered sewage wastewater-treated plants of cauliflower (Table 4). Similarly, the root length and diameter of carrot were longer in the unfiltered sewage wastewater-treated samples, while a shorter root length and diameter were measured in the filtered sewage wastewater-treated plants (Table 4).

### 3.4. Heavy Metal Buildup in Leaves

The concentration of the heavy metals was significantly higher in the leaves of the plants grown using unfiltered sewage wastewater, while significantly lower concentrations were recorded for those grown using filtered sewage wastewater during both years for the spinach, carrot, and cauliflower (Table 5). However, the Cd and Cr in the leaves of the selected vegetables were not affected by the unfiltered or filtered sewage wastewater treatments during either year of the study (Table 5).

**Table 3.** Growth, yield, and biomass of selected vegetables from unfiltered and filtered sewage wastewater.

| Sewage Wastewater | Spinach | | | Cauliflower | | | Carrot | | |
|---|---|---|---|---|---|---|---|---|---|
| | Year I | Year II | Mean | Year I | Year II | Mean | Year I | Year II | Mean |
| Number of leaves per plant | | | | | | | | | |
| Unfiltered sewage wastewater | 36.43 a | 37.36 a | 36.90 a | 30.66 a | 32.33 a | 31.50 a | 10.00 a | 9.33 a | 9.66 a |
| Filtered sewage wastewater | 29.76 a | 33.20 a | 31.48 b | 30.66 a | 27.00 a | 28.83 a | 6.97 a | 6.93 a | 6.95 b |
| Mean | 33.10 a | 35.28 a | | 30.66 a | 29.66 a | | 8.48 a | 8.13 a | |
| Fresh weight of leaves per plant (g) | | | | | | | | | |
| Unfiltered sewage wastewater | 179.34 a | 183.99 a | 181.66 a | 420.33 a | 439.33 a | 429.83 a | 56.86 a | 44.66 a | 50.76 a |
| Filtered sewage wastewater | 147.00 a | 164.72 a | 155.86 b | 409.17 a | 397.07 a | 403.12 b | 32.52 a | 34.33 a | 33.43 b |
| Mean | 163.17 a | 174.35 a | | 414.75 a | 418.20 a | | 44.69 a | 39.50 a | |
| Dry weight of leaves per plant (g) | | | | | | | | | |
| Unfiltered sewage wastewater | 14.55 a | 14.94 a | 14.74 a | 67.67 a | 70.73 a | 69.20 a | 6.20 a | 5.96 a | 6.08 a |
| Filtered sewage wastewater | 11.94 a | 13.39 a | 12.66 b | 65.87 a | 63.92 a | 64.90 a | 4.63 a | 4.86 a | 4.75 b |
| Mean | 13.25 a | 14.16 a | | 66.77 a | 67.33 a | | 5.41 a | 5.41 a | |
| Fresh weight of roots per plant (g) | | | | | | | | | |
| Unfiltered sewage wastewater | 4.30 a | 4.83 a | 4.56 a | 62.66 a | 62.50 a | 62.58 a | 101.93 a | 106.67 a | 104.30 a |
| Filtered sewage wastewater | 2.76 a | 2.33 a | 2.55 b | 54.00 a | 52.33 a | 53.16 b | 95.97 a | 100.67 a | 98.32 b |
| Mean | 3.53 a | 3.58 a | | 58.33 a | 57.41 a | | 98.95 a | 103.67 a | |
| Dry weight of roots per plant (g) | | | | | | | | | |
| Unfiltered sewage wastewater | 0.86 a | 0.96 a | 0.91 a | 16.23 a | 16.16 a | 16.20 a | 9.96 a | 10.50 a | 10.23 a |
| Filtered sewage wastewater | 0.55 a | 0.46 a | 0.51 b | 11.86 a | 12.40 a | 12.13 b | 9.03 a | 9.30 a | 9.16 b |
| Mean | 0.70 a | 0.72 a | | 14.05 a | 14.28 a | | 9.50 a | 9.90 a | |
| Biomass on fresh weight basis (g) | | | | | | | | | |
| Unfiltered sewage wastewater | 183.64 a | 188.82 a | 186.23 a | 1343.6 a | 1351.4 a | 1347.5 a | 158.80 a | 151.33 a | 155.07 a |
| Filtered sewage wastewater | 149.77 a | 167.05 a | 158.41 b | 1290.2 a | 1270.6 a | 1280.4 b | 128.49 a | 135.00 a | 131.75 b |
| Mean | 166.71 a | 177.94 a | | 1316.9 a | 1311.0 a | | 143.65 a | 143.17 a | |
| Biomass on dry weight basis (g) | | | | | | | | | |
| Unfiltered sewage wastewater | 15.41 a | 15.90 a | 15.66 a | 274.36 a | 279.08 a | 276.72 a | 16.16 a | 16.46 a | 16.31 a |
| Filtered sewage wastewater | 12.50 a | 13.85 a | 13.17 b | 264.84 a | 262.18 a | 263.51 b | 13.66 a | 14.16 a | 13.91 b |
| Mean | 13.95 a | 14.88 a | | 269.60 a | 270.63 a | | 14.91 a | 15.31 a | |

Note(s): Treatment and year's mean of three samples with different letters show significant difference at $p \leq 0.05$ (LSD test).

**Table 4.** Curd diameter, fresh and dry weights, and fresh stem weight of cauliflower and carrot from unfiltered and filtered sewage wastewater.

| Sewage Wastewater | Cauliflower | | |
|---|---|---|---|
| | Year I | Year II | Mean |
| Curd diameter (cm) | | | |
| Unfiltered sewage wastewater | 36.02 a | 36.46 a | 36.24 a |
| Filtered sewage wastewater | 31.30 a | 31.63 a | 31.46 b |
| Mean | 33.66 a | 34.05 a | |
| Curd fresh weight (g) | | | |
| Unfiltered sewage wastewater | 750.57 a | 753.23 a | 751.90 a |
| Filtered sewage wastewater | 737.53 a | 739.53 a | 738.53 b |
| Mean | 744.05 a | 746.38 a | |
| Curd dry weight (g) | | | |
| Unfiltered sewage wastewater | 182.38 a | 183.82 a | 183.10 a |
| Filtered sewage wastewater | 178.63 a | 178.62 a | 178.62 b |
| Mean | 180.51 a | 181.22 a | |
| Fresh weight of stem (g) | | | |
| Unfiltered sewage wastewater | 110.00 a | 96.33 a | 103.17 a |
| Filtered sewage wastewater | 89.50 a | 81.67 a | 85.58 b |
| Mean | 99.75 a | 89.00 a | |
| | Carrot | | |
| Root length (cm) | | | |
| Unfiltered sewage wastewater | 26.53 a | 29.00 a | 27.76 a |
| Filtered sewage wastewater | 25.23 a | 22.00 a | 23.61 b |
| Mean | 25.88 a | 25.50 a | |
| Root diameter (cm) | | | |
| Unfiltered sewage wastewater | 12.41 a | 12.93 a | 12.67 a |
| Filtered sewage wastewater | 11.06 a | 10.11 a | 10.59 b |
| Mean | 11.74 a | 11.52 a | |

Note(s): Treatment and year's mean of three samples with different letters show significant difference at $p \leq 0.05$ (LSD test).

### 3.5. Heavy Metal Concentrations in Roots

The maximum concentrations of heavy metals were recorded in the roots of plants grown using unfiltered sewage wastewater, and the minimum concentrations were found in those grown using filtered sewage wastewater (Table 6). Cd and Cr were non-significant in the treated and untreated plants of the selected vegetables (Table 6).

### 3.6. Heavy Metal Concentrations in Curds of Cauliflower

Higher concentrations of the heavy metals, i.e., $Pb^{+2}$, $Ni^{+2}$, $Cu^{+2}$, and $Fe^{+2}$, were found in the edible curd of the cauliflower plants irrigated with unfiltered sewage wastewater, while lower concentrations of these heavy metals were measured in the cauliflower plants irrigated with filtered sewage wastewater (Table 7). Both $Cd^{+2}$ and $Cr^{+2}$ were non-significant in the cauliflower plants treated and not treated with sewage wastewater during both year I and II (Table 7).

**Table 5.** Effect of unfiltered and filtered sewage wastewater on heavy metal concentrations (mg kg$^{-1}$) in leaves of spinach, cauliflower, and carrot.

| Sewage Wastewater | Spinach Leaves | | | Cauliflower Leaves | | | Carrot Leaves | | |
|---|---|---|---|---|---|---|---|---|---|
| | Year I | Year II | Mean | Year I | Year II | Mean | Year I | Year II | Mean |
| Pb$^{+2}$ content | | | | | | | | | |
| Unfiltered sewage wastewater | 4.466 a | 4.700 a | 4.583 a | 12.700 a | 12.867 a | 12.783 a | 6.400 a | 6.833 a | 6.616 a |
| Filtered sewage wastewater | 3.233 a | 3.800 a | 3.516 b | 9.400 a | 9.333 a | 9.367 b | 4.666 a | 4.833 a | 4.750 b |
| Mean | 3.850 a | 4.250 a | | 11.050 a | 11.100 a | | 5.533 a | 5.833 a | |
| Ni$^{+2}$ content | | | | | | | | | |
| Unfiltered sewage wastewater | 13.100 a | 13.300 a | 13.200 a | 13.667 a | 14.000 a | 13.833 a | 12.467 a | 13.100 a | 12.783 a |
| Filtered sewage wastewater | 10.767 a | 11.400 a | 11.083 b | 10.667 a | 10.867 a | 10.767 b | 9.867 a | 9.933 a | 9.900 b |
| Mean | 11.933 a | 12.350 a | | 12.167 a | 12.433 a | | 11.167 a | 11.517 a | |
| Cu$^{+2}$ content | | | | | | | | | |
| Unfiltered sewage wastewater | 13.800 a | 14.067 a | 13.933 a | 15.800 a | 15.633 a | 15.717 a | 11.200 a | 11.900 a | 11.550 a |
| Filtered sewage wastewater | 10.933 a | 11.200 a | 11.067 b | 12.633 a | 12.700 a | 12.667 b | 9.700 a | 9.867 a | 9.783 b |
| Mean | 12.367 a | 12.633 a | | 14.167 a | 14.217 a | | 10.450 a | 10.883 a | |
| Cd$^{+2}$ content | | | | | | | | | |
| Unfiltered sewage wastewater | 0.400 a | 0.533 a | 0.466 a | 0.433 a | 0.466 a | 0.450 a | 0.333 a | 0.466 a | 0.400 a |
| Filtered sewage wastewater | 0.090 a | 0.133 a | 0.111 a | 0.334 a | 0.334 a | 0.333 a | 0.250 a | 0.266 a | 0.258 a |
| Mean | 0.245 a | 0.333 a | | 0.383 a | 0.440 a | | 0.291 a | 0.366 a | |
| Fe$^{+2}$ content | | | | | | | | | |
| Unfiltered sewage wastewater | 294.70 a | 295.73 a | 295.22 a | 356.53 a | 360.03 a | 358.28 a | 181.00 a | 190.83 a | 185.92 a |
| Filtered sewage wastewater | 238.50 a | 246.87 a | 242.68 b | 314.30 a | 317.80 a | 316.05 b | 146.80 a | 149.77 a | 148.28 b |
| Mean | 266.60 a | 271.30 a | | 335.42 a | 338.92 a | | 163.90 a | 170.30 a | |
| Cr content | | | | | | | | | |
| Unfiltered sewage wastewater | 0.090 a | 0.126 a | 0.108 a | 2.433 a | 2.433 a | 2.433 a | 0.633 a | 0.766 a | 0.700 a |
| Filtered sewage wastewater | 0.056 a | 0.103 a | 0.079 a | 2.233 a | 2.266 a | 2.249 a | 0.466 a | 0.566 a | 0.516 a |
| Mean | 0.073 a | 0.114 a | | 2.250 a | 2.433 a | | 0.550 a | 0.666 a | |

Note(s): Treatment and year's mean of three samples with different letters show significant difference at $p \leq 0.05$ (LSD test).

**Table 6.** Effect of unfiltered and filtered sewage wastewater on heavy metal concentrations (mg kg$^{-1}$) in roots of spinach, cauliflower, and carrot.

| Sewage Wastewater | Spinach Roots | | | Cauliflower Roots | | | Carrot Roots | | |
|---|---|---|---|---|---|---|---|---|---|
| | Year I | Year II | Mean | Year I | Year II | Mean | Year I | Year II | Mean |
| Pb$^{+2}$ content | | | | | | | | | |
| Unfiltered sewage wastewater | 3.100 a | 3.166 a | 3.133 a | 6.800 a | 6.766 a | 6.783 a | 8.600 a | 8.866 a | 8.733 a |
| Filtered sewage wastewater | 2.666 a | 2.800 a | 2.733 b | 4.100 a | 4.266 a | 4.183 b | 6.800 a | 7.166 a | 6.983 b |
| Mean | 2.883 a | 2.983 a | | 5.450 a | 5.516 a | | 7.700 a | 8.016 a | |
| Ni$^{+2}$ content | | | | | | | | | |
| Unfiltered sewage wastewater | 11.133 a | 12.200 a | 11.667 a | 12.033 a | 11.733 a | 11.883 a | 15.800 a | 16.067 a | 15.933 a |
| Filtered sewage wastewater | 9.660 a | 10.133 a | 9.867 b | 9.167 a | 9.367 a | 9.267 b | 13.633 a | 14.500 a | 14.067 b |
| Mean | 10.367 a | 11.167 a | | 10.600 a | 10.550 a | | 14.717 a | 15.283 a | |
| Cu$^{+2}$ content | | | | | | | | | |
| Unfiltered sewage wastewater | 13.033 a | 13.800 a | 13.417 a | 15.467 a | 15.633 a | 15.550 a | 12.800 a | 13.800 a | 13.300 a |
| Filtered sewage wastewater | 9.267 a | 9.400 a | 9.333 b | 10.600 a | 11.100 a | 10.850 b | 9.967 a | 10.300 a | 10.133 b |
| Mean | 11.150 a | 11.600 a | | 13.033 a | 13.367 a | | 11.383 a | 12.050 a | |
| Cd$^{+2}$ content | | | | | | | | | |
| Unfiltered sewage wastewater | 0.300 a | 0.400 a | 0.350 a | 0.163 a | 0.133 a | 0.148 a | 0.600 a | 1.2667 a | 0.933 a |
| Filtered sewage wastewater | 0.090 a | 0.166 a | 0.128 a | 0.056 a | 0.023 a | 0.040 a | 0.466 a | 0.633 a | 0.550 a |
| Mean | 0.195 a | 0.283 a | | 0.110 a | 0.078 a | | 0.533 a | 0.950 a | |
| Fe$^{+2}$ content | | | | | | | | | |
| Unfiltered sewage wastewater | 258.33 a | 261.40 a | 259.87 a | 259.27 a | 261.73 a | 260.50 a | 226.53 a | 231.30 a | 228.92 a |
| Filtered sewage wastewater | 224.50 a | 233.73 a | 229.12 b | 315.47 a | 219.23 a | 217.35 b | 196.20 a | 199.37 a | 197.78 b |
| Mean | 241.42 a | 247.57 a | | 237.37 a | 240.48 a | | 211.37 a | 215.33 a | |
| Cr$^{+2}$ content | | | | | | | | | |
| Unfiltered sewage wastewater | 2.000 a | 2.100 a | 2.050 a | 2.167 a | 2.166 a | 2.166 a | 1.200 a | 1.300 a | 1.250 a |
| Filtered sewage wastewater | 1.433 a | 1.700 a | 1.566 a | 1.400 a | 1.600 a | 1.500 a | 0.833 a | 0.900 a | 0.866 a |
| Mean | 1.716 a | 1.900 a | | 1.783 a | 1.883 a | | 1.016 a | 1.100 a | |

Note(s): Treatment and year's mean of three samples with different letters show significant difference at $p \leq 0.05$ (LSD test).

**Table 7.** Effect of unfiltered and filtered sewage wastewater on heavy metal concentrations (mg kg$^{-1}$) in cauliflower curds.

| Sewage Wastewater | Cauliflower Curds | | |
|---|---|---|---|
| | **Year I** | **Year II** | **Mean** |
| Pb$^{+2}$ content | | | |
| Unfiltered sewage wastewater | 2.466 a | 2.800 a | 2.633 a |
| Filtered sewage wastewater | 3.233 a | 3.700 a | 3.466 b |
| Mean | 2.850 a | 3.250 a | |
| Ni$^{+2}$ content | | | |
| Unfiltered sewage wastewater | 11.500 a | 11.767 a | 11.633 a |
| Filtered sewage wastewater | 9.300 a | 9.733 a | 9.156 b |
| Mean | 10.400 a | 10.750 a | |
| Cu$^{+2}$ content | | | |
| Unfiltered sewage wastewater | 12.267 a | 11.867 a | 12.067 a |
| Filtered sewage wastewater | 9.033 a | 9.333 a | 9.183 b |
| Mean | 10.650 a | 10.600 a | |
| Cd$^{+2}$ content | | | |
| Unfiltered sewage wastewater | 0.060 a | 0.196 a | 0.128 a |
| Filtered sewage wastewater | 0.023 a | 0.030 a | 0.027 a |
| Mean | 0.0417 a | 0.113 a | |
| Fe$^{+2}$ content | | | |
| Unfiltered sewage wastewater | 172.83 a | 181.60 a | 177.22 a |
| Filtered sewage wastewater | 136.20 a | 138.43 a | 137.32 b |
| Mean | 154.52 a | 160.02 a | |
| Cr$^{+2}$ content | | | |
| Unfiltered sewage wastewater | 1.500 a | 1.833 a | 1.666 a |
| Filtered sewage wastewater | 0.800 a | 1.100 a | 0.950 a |
| Mean | 1.150 a | 1.466 a | |

Note(s): Treatment and year's mean of three samples with different letters show significant difference at $p \leq 0.05$ (LSD test).

## 4. Discussion

The sand-based column filter exhibited good potential to decrease the concentrations of the heavy metals found in sewage wastewater. In the present study, the sewage water filtered through a sand column filter had lower heavy metal concentrations than the unfiltered sewage wastewater. The filtration of sewage wastewater is effective for the reduction of the concentration of heavy metals. Similarly, a greater reduction of Cu$^{+2}$, Ni$^{+2}$, and Pb$^{+2}$ was reported in filtered sewage wastewater than in unfiltered sewage wastewater by Naz et al. [23]. In another study, Tauqeer et al. [24] also estimated that a sand filter had the potential to reduce the heavy metals and organic pollutants present in sewage wastewater.

The number of leaves was found to be higher in the selected vegetables irrigated with unfiltered sewage wastewater than those grown using filtered sewage wastewater. This increase in the number of leaves may possibly be due to the presence of essential nutrients present in sewage wastewater [25]. The present findings are in accordance with earlier work [26] that found that the use of sewage wastewater without filtration increased the number of leaves in okra and tomato. The fresh and dry weights of the roots, leaves, and total biomass of the selected vegetables were enhanced due to the presence of macro- and micronutrients in the sewage wastewater. A similar increase in growth and yield traits was also noted by Iqbal et al. [27] in bell peppers. Therefore, the present findings are in agreement with earlier studies, including that of Boamponsem et al. [28], who found that irrigation with sewage waste increased the fresh and dry weights of the roots, leaves, and total biomass of lettuce, carrot, and cabbage.

The fresh and dried weights of the leaves of lettuce (edible parts) increased due to sewage waste water and a similar increase in leaf-related traits were recorded in earlier work by Şentürk et al. [29]. The availability of nutrients in the sewage wastewater improved

the size and weight of carrot roots and the curd of cauliflower in the present study. Excellent leaves of lettuce, roots of carrot, and curds of cauliflower are higher yield-contributing traits. A higher yield of the studied vegetables was recorded for the unfiltered sewage wastewater-treated samples compared to those grown using filtered sewage wastewater. The decrease in the fruit yield of those plants treated with sewage wastewater is mainly due to the reduction in the mineral content during the filtration process. Hence, the present study is in agreement with the previous studies of Naz et al. [20] and Abbasi et al. [30], who found that filtered sewage wastewater contained fewer minerals than unfiltered sewage wastewater. Hence, the sand column is found to be effective for the removal of toxic organic pollutants and the reduction of heavy metals in sewage wastewater.

The results of the present study show that the heavy metal concentrations were increased in the plant leaves, roots, and edible parts due to irrigation with unfiltered sewage wastewater. The consumption of contaminated vegetables is very dangerous for health. However, a marked reduction in heavy metal concentrations was estimated to be achieved using filtered sewage wastewater. It was established that the sand column filter was capable of reducing the heavy metal concentrations in the sewage wastewater. The buildup of heavy metals increased in the edible parts due to irrigation with unfiltered sewage wastewater [31]. The sand column filter showed better potential to filter sewage wastewater for the irrigation of agricultural crops. In the present study, the reclamation of sewage wastewater did not affect the $Cd^{+2}$ and $Cr^{+2}$ levels. Different plant researchers evaluated that filtered water had a lower heavy metal concentration than unfiltered sewage wastewater [32]. The present findings are in line with the earlier work of Naz et al. [26] because they also evaluated that sand columns were capable of reducing the heavy metal concentrations in two vegetables: okra and tomato. Similarly, Naz et al. [33] also reported that heavy metals were found to be more abundant in spinach plants when irrigated with sewage wastewater. Saini et al. [34] found that sand is an effective strategy for the reduction of heavy metal concentrations from sewage wastewater that spoiled the fresh produce. Verma et al. [35] also found that sand-filter methods are effective for the elimination of heavy metals and organic pollutants from sewage wastewater, making the water fit for the irrigation of agricultural land. Chitosan-based absorbents were also used by different plant researchers for the removal of heavy metals [36–38].

## 5. Conclusions

Fresh vegetables are required for a healthy life. The amount of available freshwater is depleted due to climate change. Contaminated water may increase plant growth and yield. However, heavy metal accumulation within the fruits may increase the risk of numerous diseases for consumers/humans. The reduction of heavy metal buildup within the fruits is an essential approach. Therefore, the introduction of filtration methods is very supportive for the utilization of contaminated water for irrigation purposes. From the current results, it has been concluded that the sand column filter has good potential to reduce heavy metals, i.e., $Pb^{+2}$, $Ni^{+2}$, $Cu^{+2}$, $Fe^{+2}$, $Cd^{+2}$, and $Cr^{+2}$, from sewage water. Sand column filters are cheap, eco-friendly, non-chemical, and easy for farmers to use for filtering industrial effluents/wastewater.

**Author Contributions:** Conceptualization, supervision S.N. and M.A.A. (Muhammad Akbar Anjum); software, S.N.; analysis, R.A. and M.A.E.-S.; interpretation of data, S.N.; writing—original draft preparation, S.N.; writing—review and editing, A.S., B.S., and M.A.A. (Muhammad Ahsan Altaf); M.A.A. (Muhammad Ahsan Altaf); funding acquisition, A.S. and M.A.E.-S. All authors have read and agreed to the published version of the manuscript.

**Funding:** The authors are also thankful to the Higher Education Commission for the funding to conduct this research.

**Institutional Review Board Statement:** Not applicable.

**Informed Consent Statement:** Not applicable.

**Data Availability Statement:** The data sets used and/or analyzed in the current study are available from the corresponding author on reasonable request.

**Acknowledgments:** The authors would like to extend their sincere appreciation to the Researchers Supporting Project Number (RSP-2022/182), King Saud University, Riyadh, Saudi Arabia.

**Conflicts of Interest:** The authors declare no conflict of interest.

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
