# Peer review of "Purification of Sewage Wastewater though Sand Column Filter for Lessening of Heavy Metals Accumulation in Lettuce, Carrot, and Cauliflower"

_water, doi:10.3390/w14223770_

Round 1

Reviewer 1 Report

Suggested revisions:

  1. The manuscript generally suffers from poor grammar. It should be rewritten, with the assistance of an English language expert.
  2. Pg 3, line 122. “…by spreading on clean place for 60 mint.” Is it “mint” or minutes?
  3. Tables 2-7 have some values labeled as “a” and others as “b”. What do these labels stand for? They need to be defined immediately after the tables.
  4. Pg 4, lines 152 & 153. “According to *WHO/FAO, all the studied heavy metals …”; a reference is required since none has also been provided with respect to the WHO/FAO values in Table 2.
  5. Pg 5, Table 3. “Number of leave per plant (g)”; are the units appropriate as (g) since that it’s referring to a number?  
  6. Pg 9, Cr content. The means of the Cr content in the spinach and cauliflower leaves, for the unfiltered sewage wastewater, filtered sewage wastewater, and the yearly ones, are not correct.

7.     Pg 10, Pb and Ni content. The means of the Pb content in the cauliflower curds, for the unfiltered sewage wastewater and filtered sewage wastewater, are not correct. The means of the Ni content in the cauliflower curds, for the unfiltered sewage wastewater, filtered sewage wastewater, and the yearly ones, are not correct.

Author Response

Dear Editor

We are pleased to re-submit our revised manuscript, “Purification of Sewage Wastewater Though Sand Column Filter for Lessening of Heavy Metals Accumulation in Lettuce, Carrot and Cauliflower” Ms. Ref. No.: Water-2012542) to be considered for publication in “Water.

Thank you very much for making a critical assessment of the original version of our manuscript. We are glad to learn that the editor as well as the reviewer(s) considered our findings meaningful. However, the reviewer(s) provided some valuable comments and suggestions for the potential improvement of the manuscript. Based on the comments, you encouraged us to re-submit a revised version, which takes into account all of the points raised by the reviewer(s). We have now completed a thorough revision as per the recommendations of the reviewer(s). Revised portions are marked in red in the paper. Our response to reviewers’ comments has been listed below point-by-point.

Reviewer 1

The manuscript generally suffers from poor grammar. It should be rewritten, with the assistance of an English language expert.

Response back: English grammar is thoroughly corrected by Horticulture expert. Whole manuscript is thoroughly studied and edited. Corrections are highlighted with green color. Now, it will be good for readers.

Pg 3, line 122. “…by spreading on clean place for 60 mint.” Is it “mint” or minutes?

Response back: Corrected as per suggestions. Mint has been replaced with minutes

Tables 2-7 have some values labeled as “a” and others as “b”. What do these labels stand for? They need to be defined immediately after the tables.

Response back: Below the tables, lines are added.  Letters showed the maximum (a) and minimum value (b). Detailed line is added below the table and mentioned in red color.

Pg 4, lines 152 & 153. “According to *WHO/FAO, all the studied heavy metals …”; a reference is required since none has also been provided with respect to the WHO/FAO values in Table 2.

Response back: A reference has been incorporated in the manuscript.

Pg 5, Table 3. “Number of leave per plant (g)”; are the units appropriate as (g) since that it’s referring to a number?  

Response back: Correction has been made as per suggestion of reviewer. Thanks to reviewer however, it was a topographical error.

Pg 9, Cr content. The means of the Cr content in the spinach and cauliflower leaves, for the unfiltered sewage wastewater, filtered sewage wastewater, and the yearly ones, are not correct.

Response back: Value of Cr content are corrected in the manuscript and highlighted with red color.

Pg 10, Pb and Ni content. The means of the Pb content in the cauliflower curds, for the unfiltered sewage wastewater and filtered sewage wastewater, are not correct. The means of the Ni content in the cauliflower curds, for the unfiltered sewage wastewater, filtered sewage wastewater, and the yearly ones, are not correct.

Response back: Means have been thoroughly checked and corrected.

Reviewer 2 Report

  1. The English of the text should be checked
  2. The authors must be included new, relevant and more information about other adsorbent materials. Also, must be included more advantages and disadvantage of adsorption in comparison with other natural polymer. The following references can be included in the Introduction part to improve the quality of manuscript, because they provide relevant information:

https://doi.org/10.1016/j.arabjc.2021.103543;

 https://doi.org/10.3390/polym14061107;

 https://doi.org/10.3390/ijerph191711123

3.       Heavy metals in sewage should be metal ions, metal elements in the text should be changed into ions, and indicate the valence state of ions.

4.       When averaging the data in the table, the number of samples or parallel experimental tests should be provided.

Author Response

Dear Editor

We are pleased to re-submit our revised manuscript, “Purification of Sewage Wastewater Though Sand Column Filter for Lessening of Heavy Metals Accumulation in Lettuce, Carrot and Cauliflower” Ms. Ref. No.: Water-2012542) to be considered for publication in “Water.

Thank you very much for making a critical assessment of the original version of our manuscript. We are glad to learn that the editor as well as the reviewer(s) considered our findings meaningful. However, the reviewer(s) provided some valuable comments and suggestions for the potential improvement of the manuscript. Based on the comments, you encouraged us to re-submit a revised version, which takes into account all of the points raised by the reviewer(s). We have now completed a thorough revision as per the recommendations of the reviewer(s). Revised portions are marked in red in the paper. Our response to reviewers’ comments has been listed below point-by-point

Reviewer 2

The English of the text should be checked

Response back: Checked thoroughly as per suggestions. Some mistakes were observed which are corrected.

The authors must be included new, relevant and more information about other adsorbent materials. Also, must be included more advantages and disadvantage of adsorption in comparison with other natural polymer. The following references can be included in the Introduction part to improve the quality of manuscript, because they provide relevant information:

https://doi.org/10.1016/j.arabjc.2021.103543;

 https://doi.org/10.3390/polym14061107;

 https://doi.org/10.3390/ijerph191711123

Response back: Related references have been added within the manuscript

Heavy metals in sewage should be metal ions, metal elements in the text should be changed into ions, and indicate the valence state of ions.

Response back: Metal ions has been changes into ions as per suggestions and valency of ions is also added in the whole manuscript where is required.

When averaging the data in the table, the number of samples or parallel experimental tests should be provided.

Response back: A clear statement below the tables is listed to clear the number of samples.

Reviewer 3 Report

Specific comments

1.      The author should highlight the results in the abstract.

2.      Please include basic water parameters such as TDS and Hardness of water.

3.      The author includes the health effects points in the introduction part.

4.      Conclusions need more points and novelty.

5.      Grammar and typos. The manuscript contains some grammatical and typographical errors. The authors need to thoroughly revise the manuscript and correct the errors.

6.      Moreover, there are a few corrections that need to be adopted for the overall presentation of this manuscript.

Author Response

Dear Editor

We are pleased to re-submit our revised manuscript, “Purification of Sewage Wastewater Though Sand Column Filter for Lessening of Heavy Metals Accumulation in Lettuce, Carrot and Cauliflower” Ms. Ref. No.: Water-2012542) to be considered for publication in “Water.

Thank you very much for making a critical assessment of the original version of our manuscript. We are glad to learn that the editor as well as the reviewer(s) considered our findings meaningful. However, the reviewer(s) provided some valuable comments and suggestions for the potential improvement of the manuscript. Based on the comments, you encouraged us to re-submit a revised version, which takes into account all of the points raised by the reviewer(s). We have now completed a thorough revision as per the recommendations of the reviewer(s). Revised portions are marked in red in the paper. Our response to reviewers’ comments has been listed below point-by-point

Reviewer 3

The author should highlight the results in the abstract.

Response back: Results are added in the abstract.

Please include basic water parameters such as TDS and Hardness of water.

Response back: TDS and Hardness of water have been included in the manuscript

The author includes the health effects points in the introduction part.

Response back: Detailed health effects points are incorporated within the manuscript.

Conclusions need more points and novelty.

Response back: More points are incorporated in the conclusion.

Grammar and typos. The manuscript contains some grammatical and typographical errors. The authors need to thoroughly revise the manuscript and correct the errors.

Response back: Manuscript has been revised by frequent English speaker.

 Moreover, there are a few corrections that need to be adopted for the overall presentation of this manuscript.

Response back: All the listed comments have been briefly described in the manuscript with red color.

Round 2

Reviewer 1 Report

Generally, there has been good attempt in responding to the queries.

Reviewer 2 Report

 Accept in present form